# The Gastrointestinal Tract Is Pinpointed as a Reservoir of *Candida albicans*, *Candida parapsilosis*, and *Candida tropicalis* Genotypes Found in Blood and Intra-Abdominal Samples

**DOI:** 10.3390/jof9070732

**Published:** 2023-07-06

**Authors:** Aina Mesquida, Marina Machado, Lorena Dávila-Cherres, Teresa Vicente, Carlos Sánchez-Carrillo, Luis Alcalá, Elena Reigadas, Patricia Muñoz, Jesús Guinea, Pilar Escribano

**Affiliations:** 1Clinical Microbiology and Infectious Diseases Department, Hospital General Universitario Gregorio Marañón, C/Dr. Esquerdo, 46, 28007 Madrid, Spain; ainacmesquida@gmail.com (A.M.); marinamachadov@gmail.com (M.M.); loreadavila20@gmail.com (L.D.-C.); mtvicente75@hotmail.com (T.V.); sanchezcarrillocarlos@gmail.com (C.S.-C.); uisalcala@efd.net (L.A.); helenrei@hotmail.com (E.R.); pmunoz@hggm.es (P.M.); 2Instituto de Investigación Sanitaria Gregorio Marañón, 28007 Madrid, Spain; 3CIBER Enfermedades Respiratorias-CIBERES (CB06/06/0058), 28029 Madrid, Spain; 4Department of Medicine, School of Medicine, Universidad Complutense de Madrid, 28040 Madrid, Spain; 5School of Health Sciences-HM Hospitals, Camilo José Cela University, 28692 Madrid, Spain

**Keywords:** genotyping, *Candida*, candidaemia, gastro-intestinal tract

## Abstract

Background: *Candida* spp., as part of the microbiota, can colonise the gastrointestinal tract. We hypothesised that genotyping *Candida* spp. isolates from the gastrointestinal tract could help spot genotypes able to cause invasive infections. Materials/methods: A total of 816 isolates of *C. albicans* (n = 595), *C. parapsilosis* (n = 118), and *C. tropicalis* (n = 103) from rectal swabs (n = 754 patients) were studied. Genotyping was conducted using species-specific microsatellite markers. Rectal swab genotypes were compared with previously studied blood (n = 814) and intra-abdominal (n = 202) genotypes. Results: A total of 36/754 patients had the same *Candida* spp. isolated from blood cultures, intra-abdominal samples, or both; these patients had candidemia (n = 18), intra-abdominal candidiasis (n = 11), both clinical forms (n = 1), and non-significant isolation (n = 6). Genotypes matching the rectal swab and their blood cultures (84.2%) or their intra-abdominal samples (92.3%) were found in most of the significant patients. We detected 656 genotypes from rectal swabs, 88.4% of which were singletons and 11.6% were clusters. Of these 656 rectal swab genotypes, 94 (14.3%) were also detected in blood cultures and 34 (5.2%) in intra-abdominal samples. Of the rectal swab clusters, 62.7% were previously defined as a widespread genotype. Conclusions: Our study pinpoints the gastrointestinal tract as a potential reservoir of potentially invasive *Candida* spp. genotypes.

## 1. Introduction

*Candida* spp. forms part of the gastrointestinal microbiota, especially dominated by *Candida albicans* and *Candida tropicalis* and, to a lesser extent, *Candida parapsilosis* and *Candida glabrata* [1,2,3,4].

*Candida* spp. genotyping may play a key role in revealing the source of invasive infections. We previously proved the catheter to be an exogenous source of infections in patients with catheter-related candidemia [5,6]. Other compartments, such as the vaginal cavity, do not contain genotypes that can be detected in blood cultures [7]. Alternatively, infection could be caused by the translocation of endogenous *Candida* spp. strains colonising the gastrointestinal tract, although the relationship among *Candida* spp. gastrointestinal isolates and bloodstream infection is unclear. Genotyping data from *Candida* spp. isolates from gastrointestinal tract samples are scarce, reported 20 years ago, and generated using low-resolution genotyping techniques [8,9,10,11]. One study has recently tried to trace the gastrointestinal origin of bloodstream infection isolates using a highly discriminative technique. However, the low number of patients for whom paired faecal and blood samples were available only demonstrated the hypothesis in two patients with candidemia caused by *C. parapsilosis* [12].

We hypothesised that genotyping *Candida* spp. isolates from the gastrointestinal tract could help spot genotypes able to cause invasive infections. The aim of this study was to tag the gastrointestinal tract as a potential reservoir of *C. albicans*, *C. parapsilosis*, and *C. tropicalis* genotypes that could potentially be present in blood culture and intra-abdominal samples.

## 2. Materials and Methods

### 2.1. Isolates Studied

From February 2019 to August 2022, *Candida* spp. isolates were prospectively collected (n = 816; *C. albicans,* n = 595; *C. parapsilosis*, n = 118; and *C. tropicalis,* n = 103) from rectal swabs of patients (n = 754) admitted to the Gregorio Marañón Hospital, Madrid, Spain; samples were screened for the presence of multi-resistant bacteria (such as carbapenemase- or BLEA-producing *Enterobacteriaceae*, *Pseudomonas aeruginosa*, vancomycin-resistant enterococci, and *Acinetobacter* spp.) and were positive for *Candida* spp. on ChromAgar^®^ agar plates. Such screening is conducted in patients newly admitted to an intensive care unit (and on a weekly basis thereafter), in patients previously infected/colonised by multi-resistant isolates, or in patients admitted close to or in a room where a colonised/infected patient was previously located. We studied one rectal swab sample per patient and species. Since we proved that gastrointestinal tract *Candida* spp. colonisation was mostly monoclonal, a single colony per species and sample was subcultured and stored [13].

### 2.2. Microsatellite Genotyping

Genotyping was conducted directly from cultures avoiding DNA extraction as previously reported [7,14]. Species-specific microsatellite markers were used to genotype isolates of *C. albicans* (CDC3, EF3, HIS3 CAI, CAIII, and CAVI) [15,16], *C. parapsilosis* (CP1, CP4a, CP6, and B) [17,18], and *C. tropicalis* (Ctrm1, Ctrm10, Ctrm12, Ctrm21, Ctrm24, and Ctrm28) [19]. The isolates were considered to have identical genotypes when they presented the same alleles at all markers studied. We adopted some definitions from previous studies. Briefly, singleton was defined as a genotype found in a single given patient; cluster as an identical genotype found in samples from ≥2 given patients; and patients involved in each cluster were considered epidemiologically linked if they were admitted to the same ward within a period of 12 months. Finally, a match was an identical or clonally related (a difference at only one microsatellite locus) genotype found in different samples from a given patient [7,14].

### 2.3. Data Analysis and Population Structure

Genotypes found in rectal swabs were compared with those retrieved from our database and previously detected in blood cultures (814 isolates/650 genotypes) and intra-abdominal samples (201 isolates/179 genotypes) from patients admitted to our hospital from 2007 to 2022. Matches and clusters were classified according to their isolate source: exclusively from rectal swabs, intra-abdominal samples, blood cultures, or a combination of these (rectal–abdominal, rectal–blood, rectal–abdominal–blood). We assessed how many clusters were widespread genotypes (clusters involving patients admitted to different hospitals) coming from previous studies [14,20,21]. The percentage of clusters and proportion of patients involved in clusters were compared using a standard binomial method (95% confidence intervals) (Epidat 3.1 software, Servicio de Información sobre Saúde Pública de la Dirección Xeral de Saúde Pública de la Consellería de Sanidade, Xunta de Galicia, Spain).

Population structure was analysed using Structure software version 2.3.4 [22]. Independent runs were performed by using K (number of populations) from one to ten populations, a burn-in of 10,000 Markov chain Monte Carlo (MCMC) iterations, and a data collection period of 100,000 MCMC iterations. Each simulation of K was replicated 10 times. To estimate the most likely K, we used Structure Harvester [23]. Structure composition estimates the log probability of the data for each value of K. Finally, population differentiation was studied via analysis of Molecular Variance (AMOVA) using GenAlex v 6.51 software [24].

### 2.4. Clinical Data

Demographic and clinical data of patients who had the same *Candida* spp. isolated from rectal swabs and blood cultures and/or intra-abdominal samples were collected from clinical records and included clinical significance of the isolation, risk factors for developing invasive candidiasis, prior antifungal use at the time or within three months before the isolation, clinical infection form, and outcome.

## 3. Results

### 3.1. Intra-Patient Analysis of Genotypes Found among Rectal Swabs, Blood Cultures, and Intra-Abdominal Samples

A total of 36/754 patients also had the same *Candida* spp. isolated from blood cultures (n = 17), intra-abdominal samples (n = 17), or both (n = 2) within the same period of time. The patients had candidemia (n = 18), intra-abdominal candidiasis (n = 11), both clinical forms (n = 1), and non-significant isolation (n = 6). The patients with invasive candidiasis (n = 30) are described in Table 1 and Appendix A.

Of the 30 patients with *Candida* spp. in rectal swabs and isolates from blood (n = 19) and/or intra-abdominal samples (n = 13), 16/19 (84.2%) and 12/13 (92.3%) patients had genotypes matching the rectal swab and their blood cultures or their intra-abdominal samples, respectively. Matches were distributed among identical genotypes in 20 patients or clonally related genotypes in 8 patients; the latter were frequently rectal–abdominal matches (Table 2). With the exception of two patients who had candidemia caused by *C. tropicalis*, clonally related genotypes were found in patients with intra-abdominal candidiasis.

*C. albicans* was the species contributing the highest number of isolates, thus resulting in the highest number of matches, and the proportions of rectal–blood matches and rectal–abdominal matches were similar. We had isolates available from the three sample types in two patients; a genotype matching the three sample types was found in one of them (patient P02), whereas in the remaining patient (patient P01) we found two different genotypes; one genotype matched the rectal swab and the intra-abdominal sample, and an unrelated genotype was found in the blood culture that was identical to a successive rectal swab genotype (Figure 1 and Appendix A).

The remaining patients presented unrelated genotypes from rectal swabs and other sample types (n = 4) (Table 2 and Appendix A). For more details about the patients with non-significant isolation (n = 6), see Appendix A.

In these patients with the same *Candida* spp. isolated from blood cultures, intra-abdominal samples, or both, whose isolations were considered as clinically significant, all available additional rectal swabs were studied. We had 19 multiple rectal swab isolates (1–7 additional isolates per patient) from 10/30 patients, distributed as *C. albicans* (n = 1 patient with isolates from the three sample types; n = 2 patients with isolates from rectal swabs and blood cultures; n = 4 patients with isolates from rectal swabs and intra-abdominal samples), *C. tropicalis* (n = 2 patients with isolates from rectal swabs and blood cultures), and *C. parapsilosis* (a patient with isolates from rectal swabs and blood cultures) (Figure 1). In 7/10 patients, genotypes matched the rectal swab samples collected at different time points; the remaining 2/10 patients presented two unrelated genotypes (patients P01 and P30) or three unrelated genotypes (patient P27) (Figure 1).

### 3.2. Rectal Swab Genotypes

Overall, we found 656 genotypes from rectal swabs (*C. albicans* [n = 466], *C. parapsilosis* [n = 103], and *C. tropicalis* [n = 87]); 88.4% of these were singletons and 11.6% were clusters (Appendix A). Differences in the proportions of clusters among species (*C. albicans*, 12.2%; *C. parapsilosis*, 11.7%; and *C. tropicalis*, 8.1%) did not reach statistical significance (*p* > 0.05). A total of 236 patients (31.3%) were involved in clusters (2 to 18 patients per cluster), with *C. albicans* being the species involved in the highest number of patients. However, differences among species did not reach statistical significance (*p* = 0.069; Table 3). Of the 236 patients involved in clusters, 30 (12.7%) patients involved in 10 clusters (2–5 patients per cluster) were epidemiologically linked and admitted to intensive care (n = 12/30), internal medicine (n = 6/30), and other wards (n = 12/30).

### 3.3. Interpatient Analysis of Genotypes Found in Rectal Swabs, Blood Cultures, and Intra-Abdominal Samples. Population Structure

Genotypes found in rectal swabs (n = 656) were compared with those retrieved from our database and sourced from blood cultures (n = 650) or intra-abdominal samples (n = 179) (Appendix A). The distribution of genotypes among species and sample types are shown in Table 4. Whereas we did not find differences reaching statistical significance in the proportion of clusters/patients involving exclusively either rectal swabs (11.6%/31.3%) or blood cultures (11.2%/29.6%), these proportions were significantly lower than in intra-abdominal samples clusters (6.2%/17.5%; *p* < 0.05), and never involved *C. parapsilosis* (Table 3 and Table 4).

Of all the genotypes found in rectal swabs, 102 were clusters distributed between rectal–blood clusters (n = 68; 10.4%), rectal–abdominal clusters (n = 8; 1.2%), and rectal–abdominal–blood clusters (n = 26; 4%); therefore, 94 (14.3%) and 34 (5.2%) rectal swab genotypes were also detected either in blood cultures or intra-abdominal samples, respectively (Table 5). Differences among species did not reach statistical significance.

We found 58/102 (56.8%) rectal swab genotypes also found in blood cultures and/or intra-abdominal samples that were widespread genotypes distributed as rectal–blood clusters (n = 28; 43.8%), rectal–abdominal (n = 6; 9.4%), and rectal–abdominal–blood clusters (n = 24; 37.5%) (Figure 2). Interestingly, the higher the probability of a cluster involving both rectal swabs and blood cultures, the higher the odds the cluster was widespread; this observation was similar among species (*p* > 0.05).

Per species population structure differed and was supported by the different K-value detected in each species (*C. albicans* [K = 4]; *C. parapsilosis* [K = 5], and *C. tropicalis* [K = 2]). No population structure was detected and AMOVA analysis showed no molecular variation between any type of sample and supporting the lack of population structure.

## 4. Discussion

Our study pinpoints the gastrointestinal tract as a potential reservoir of *Candida* spp. potentially causing candidaemia. It is supported by the set of patients who had isolations from rectal swabs and candidemia and/or intra-abdominal candidiasis in whom genotypes from rectal swabs matched the ones found in profound samples. Moreover, in the multiple rectal swabs available from these patients, a given genotype could be detected in samples taken at different time points and matched the blood culture and/or intra-abdominal sample isolates. Furthermore, 14% of genotypes found in rectal swab samples could also be detected in blood cultures. A notable proportion of genotypes (11.6%) were found in rectal swabs from different patients (clusters), with this proportion being similar to that found for genotypes from blood cultures. Moreover, the higher the probability of a cluster involving rectal swabs and blood cultures, the higher the odds the cluster was widespread.

Ours is the first study to assess *Candida* spp. genotypes detected in a large number of isolates (and patients) from rectal swab samples. We assumed that genotypes found in the rectum should be those able to translocate to precedent sections of the small or large intestine. In fact, we found identical genotypes in the rectum and causing infections in the 84.5% and 92.3% of patients with candidemia and/or intra-abdominal candidiasis, respectively. In the remaining patients (patients P09, P25, and P28), the genotype found in rectal swabs was different from that causing invasive infection; however, in these patients only one rectal isolate was available. The presence of polyclonality in the rectum in these patients cannot be ruled out [13], and additional rectal swab isolates could be matched with the invasive genotype as we observed in patient P1.

In agreement with our prior studies, we found that the percentage of clusters in rectal swabs (11.6%) was similar to that of blood cultures [14,25,26]. Interestingly, 14% of genotypes from rectal swab samples could also be detected in blood cultures. However, this proportion reached 92% when those genotypes were clusters. Nevertheless, the proportion of clusters was significantly lower in intra-abdominal isolates (6%) and was never involved *C. parapsilosis*; in fact, only one-third of rectal swab clusters could also be detected in intra-abdominal samples. That could be a consequence of the lower number of isolates available from that anatomical cavity, or the lower predisposition of intestinal *Candida* genotypes to cause intra-abdominal infections.

The presence of clusters involving epidemiologically unrelated patients is not new, as we previously reported, and is sometimes as widespread as genotypes grouping patients from different cities or countries [14,21]. We found some rectal swab clusters (66%) were widespread and involved patients in the absence of epidemiological links; the interpretation of widespread genotypes is tricky and might suggest that certain clones are spread worldwide and could be found colonising a large number of healthy subjects [14,20,21]. One of the main observations across species here reported is that the higher the probability of a cluster involving rectal swabs and blood cultures, the higher the odds the cluster is widespread. Therefore, the presence of identical genotypes in rectal swabs and blood cultures from different patients might pinpoint some saprophytic genotypes with the ability to cause invasive disease. We think that rectal swab clusters are genotypes with the ability to cause candidaemia, therefore supporting the hypothesis that the gut might be the reservoir of these genotypes. Such a conclusion was reinforced by the finding that there was no population structure or molecular variation between different anatomical sites. In fact, *C. albicans* genotypes from vaginal samples and blood cultures presented genotypically different populations, suggesting that isolates from vaginal samples are not prone to cause bloodstream infections [7].

Different scores showed that prior colonisation is a major risk factor for acquiring invasive candidiasis [27], and some studies reported that identical or clonally related genotypes can be found simultaneously colonising patients and causing candidaemia [10,28]. Previous studies in which isolates from rectal swabs and blood cultures were genotyped are somewhat dated and were conducted using a limited number of isolates [8,9,10]. Our study was not tailored to study *Candida* genotypes colonising patients and able to cause candidemia as well; however, we took advantage of some patients from whom we had multiple swab and blood culture/intra-abdominal samples available. In most patients, the genotype found in rectal swabs was also detected in blood cultures and/or intra-abdominal samples. Such observations agree with the interpatient analysis and we highlight the gut as a reservoir of genotypes able to reach the bloodstream.

Despite the fact that we performed genotyping with microsatellite markers, a highly discriminative tool, we cannot rule out higher discrimination if whole-genome sequencing had been carried out. However, studies comparing microsatellite genotyping and whole-genome sequencing in *Candida* spp. showed than the genotyping data were mostly correlated [29,30,31].

Our study is subject to some limitations. *C. albicans* outnumbered the other species; thus, the observations reported herein are skewed by the large number of isolates from this species. However, it is not surprising considering that *C. albicans* is responsible for no less than 50% of candidaemia cases and is also the dominant *Candida* spp. in the gut microbiome [1,32]. The number of isolates from intra-abdominal samples was much lower than those from rectal swabs and blood cultures; this might explain why we found a lower proportion of clusters involving intra-abdominal samples and rectal swabs; future studies including larger numbers of isolates from that cavity are warranted. Finally, genotypes in 18 patients with paired rectal swabs, and/or blood cultures, or intra-abdominal samples genotypes were first detected in blood cultures/intra-abdominal samples. Given that our study was not addressed to detect rectal swab genotypes from the very moment of a patient’s hospital admission, we cannot rule in/out that the rectal swab genotypes might not have been detected prior to the development of the infection. However, in patients with multiple rectal swab genotypes over time, the genotypes found were always identical.

## 5. Conclusions

In conclusion, our study pinpoints the gastrointestinal tract as a potential reservoir of *Candida* spp. able to cause candidaemia. Genotypes found in rectal swabs are commonly found in different patients. Finally, when rectal swab clusters also involved blood cultures, the cluster tended to be widespread.

## Figures and Tables

**Figure 1 jof-09-00732-f001:**
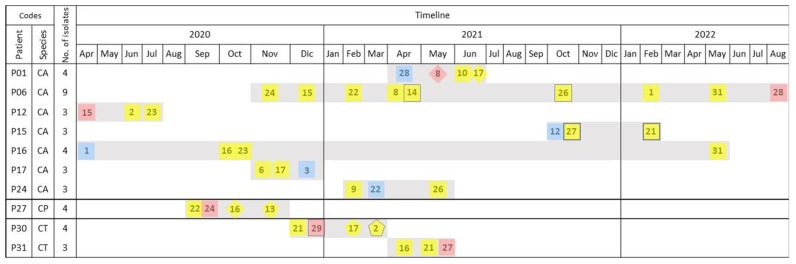
Timeline showing rectal swab genotypes from the 10 patients with multiple rectal swab, blood culture, and/or intra-abdominal isolates available. CA, *C. albicans*; CP, *C. parapsilosis*; CT, *C. tropicalis*. Symbol colours indicate the isolate’s clinical source (rectal swabs are depicted in yellow; blood cultures are depicted in red; intra-abdominal samples are depicted in blue). In a given patient, an identical symbol shape indicates matches (identical genotype) and solid-bold edges depict clonally related genotypes. The number within the symbols indicates the day of sample collection. Grey-coloured cells indicate the latency period (time spanning two points in which isolates were detected).

**Figure 2 jof-09-00732-f002:**
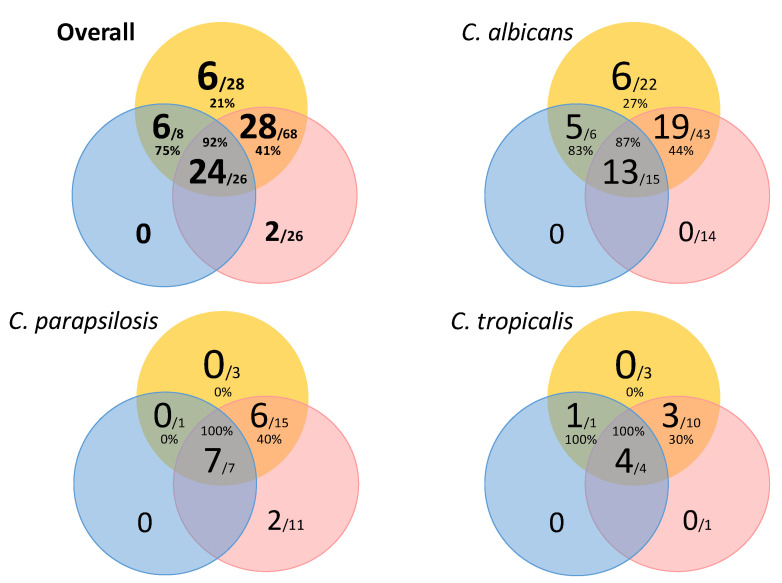
Venn diagram showing the interpatient analysis alongside widespread clusters involving rectal swabs, and/or blood culture, and/or intra-abdominal samples, broken down by species. Circle colours indicate the isolate’s clinical source (rectal swabs are depicted in yellow; blood cultures are depicted in red; intra-abdominal samples are depicted in blue). Numbers in the largest font indicate widespread clusters; numbers in the smallest font indicate total clusters found. Percentages indicate proportions of widespread clusters.

**Table 1 jof-09-00732-t001:** Description of the 30 patients with the same *Candida* spp. isolated from blood cultures, intra-abdominal samples, or both, whose isolations were considered as clinically significant.

	No. (%)
Sex (male)	22 (73.3%)
Age (mean, IQR)	66.6 (75–59)
Antibiotic use	27 (90%)
Catheter use	23 (76.6%)
Total parenteral nutrition	22 (73.3%)
Prior *Candida* colonisation	14 (46.6%)
Prior corticosteroid use	16 (53.3%)
Prior surgery	13 (43.3%)
Prior CMV infection	4 (13.3%)
ICU admission	17 (56.6%)
Haemodialysis	2 (6.6%)
Chemotherapy	2 (6.6%)
Prior antifungal use	
Azoles	7 (23.3%)
Echinocandins	3 (10%)
Both	2 (6.6%)
*Species distribution*	
*C. albicans*	19 (63.3%)
*C. parapsilosis*	5 (16.6%)
*C. tropicalis*	6 (20%)
Source of the infection	
Primary	2 (6.6%)
Catheter-related infection	15 (50%)
Intra-abdominal infection	12 (40%)
Other	1 (3.3%)
Outcome (poor)	12 (40%)

**Table 2 jof-09-00732-t002:** Intra-patient analysis of the 30 patients with rectal swab isolates and additional isolates from blood and/or intra-abdominal samples who had invasive infections: analysis of matches (identical or clonally related genotypes) and unrelated genotypes.

Isolate Source	*C. albicans* (No. Patients) *			*C. parapsilosis* (No. Patients)			*C. tropicalis* (No. Patients)			Overall (No. Patients) *		
	Identical	Clonally Related	Unrelated	Identical	Clonally Related	Unrelated	Identical	Clonally Related	Unrelated	Identical	Clonally Related	Unrelated
Rectal swab + Blood culture	10	0	2	2	0	1	2	2	0	14	2	3
Rectal swab + Intra-abdominal sample	5	4	0	0	1	1	1	1	0	6	6	1

* Two patients had *C. albicans* isolates from the three sample types.

**Table 3 jof-09-00732-t003:** Rectal swab isolates and genotyping.

	*C. albicans*	*C. parapsilosis*	*C. tropicalis*	Overall
Number of isolates/patients	595	118	103	816/754
Number of genotypes	466	103	87	656
Number of singletons	409	91	80	580
Number of clusters (%)	57 (12.2%)	12 (11.7%)	7 (8.1%)	76 (11.6%)
% (range) of patients involved in clusters	31.3% (2–18)	22.9% (2–4)	22.3% (2–7)	31.3% (2–18)

**Table 4 jof-09-00732-t004:** Blood culture and intra-abdominal sample isolates and genotyping.

	*C. albicans*		*C. parapsilosis*		*C. tropicalis*		Overall	
	Blood Culture	Intra-Abdominal Samples	Blood Culture	Intra-Abdominal Samples	Blood Culture	Intra-Abdominal Samples	Blood Culture	Intra-Abdominal Samples
Number of isolates/patients	479	158	256	22	79	21	814/800	201/189
Number of genotypes	398	137	181	22	71	20	650	179
Number of singletons	354	127	158	22	65	19	577	168
Number of clusters (%)	44 (11.1%)	10 (7.3%)	23 (12.7%)	0	6 (8.5%)	1 (5%)	73 (11.2%)	11 (6.2%)
% (range) of patients involved in clusters	26.1% (2–9)	19.6% (2–8)	38.3% (2–18)	0%	17.7% (2–3)	9.5% (2)	29.6% (2–18)	17.5% (2–8)

**Table 5 jof-09-00732-t005:** Interpatient analysis of genotypes from rectal swabs also found in blood cultures and/or intra-abdominal samples.

Species	Rectal Swab Genotypes (No.)	Rectal Swab Genotypes Found in (No. and %)			
		Blood Cultures *	Intra-Abdominal Samples *	Blood Culture and Intra-Abdominal Samples	Overall
*C. albicans*	466	43 (9.2%)	6 (1.3%)	15 (3.2%)	64 (13.7%)
*C. parapsilosis*	103	15 (14.6%)	1 (1%)	7 (6.8%)	23 (22.3%)
*C. tropicalis*	87	10 (11.5%)	1 (1.1%)	4 (4.6%)	15 (17.2%)
Overall	656	68 (10.4%)	8 (1.2%)	26 (4%)	102 (15.5%)

* Genotype(s) found exclusively in blood cultures or intra-abdominal samples.

## Data Availability

Not applicable.

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
