# Peer review of "The Gastrointestinal Tract Is Pinpointed as a Reservoir of Candida albicans, Candida parapsilosis, and Candida tropicalis Genotypes Found in Blood and Intra-Abdominal Samples"

_jof, 2023, doi:10.3390/jof9070732_

Round 1

Reviewer 1 Report

The authors performed an interesting genotyping analysis on three Candida spp from rectal swabs, blood and abdomen and found clustering of these isolates between these sample types between different patients, but also within patients.  

Major comments:

The authors draw the conclusion that the gastrointestinal tract might constitute a potential reservoir of Candida spp potentially causing candidaemia as 14% of genotypes found in rectal swabs samples could also be detected in blood cultures. As different patient groups were used for this comparison, I do not agree with the interpretation of this data. In my opinion this is likely due to common environmental strains (in the study with vaginal swabs in general also much less clusters were found, so it is logical that as a consequence also less clusters with both vaginal and blood isolates are found; this study also contained much less isolates). However, the authors also found that “a total of 41/754 patients also had the same Candida spp isolated from blood cultures (n=18), intra-abdominal samples (n=20), or both (n=3). From these 41 patients with Candida spp in rectal swabs and isolates from blood (n=21) and/or  intra-abdominal samples (n=23), 17/21 (80.9%) and 18/23 (78.3%) patients had genotypes matching the rectal swab and their blood cultures or their intra-abdominal samples, respectively.” I think this is much stronger evidence that “the gastrointestinal tract might constitute a potential reservoir of Candida spp potentially causing candidaemia”. These observed matches are  not based on coincidence as between patients this number is much lower (14% on average). I think the manuscript should focus on the latter finding and use the other data as a “control group”.   

Minor comments:

1) Raw STR data should be provided as supplementary data.

2) Table 1 and 2: Include the number of patients in both tables. Without this information it is hard to understand the percentages at the bottom of the tables. 

3) Lines 118-120: Include description regarding isolates from database (blood cultures (n=650) or intra-abdominal samples (n=150)) in M&M. Also the structure of this paragraph should be improved. First describe Table 2 and then compare R, B and A. 

4) Figure 1/lines 138-141; description colors and numbers should be in Figure legend. Improve explanation of Figure 1; in lines 126-129 and 142-144 is referred to Figure 1, but I cannot find these percentages in this figure.

5) Figure 2/lines 154-155: description colors and numbers should be in Figure legend.

6) Figure 3/lines 188-193: description colors and numbers should be in Figure legend. Indicate what the number within the symbols represent (day of month?). Dotted lines are shown in different thickness. Is there any difference?

7) Lines 59-62: The sentence “A total 59 of 41/754 patients also had the same Candida spp isolated from blood cultures (n=18), intra-60 abdominal samples (n=20), or both (n=3).” should be in the Results section (line 162). Rephrase M&M like “In patients in whom the same Candida spp was isolated from gastrointestinal tract and blood and/or abdomen, we studied available additional 61 rectal swabs, blood cultures, and intra-abdominal isolates.”

8) Lines 111-114: I think this sentence “Of the 236 patients involved in clusters, 30 (12.7%) were involved in 10 clusters (2-5 patients per cluster) and were epidemiologically linked and admitted to intensive care (n=12/30), internal medicine (n=6/30), and other wards (n=12/30).” should read as “Of the 236 patients involved in clusters, 30 (12.7%) patients, involved in 10 clusters (2-5 patients per cluster), were epidemiologically linked and admitted to intensive care (n=12/30), internal medicine (n=6/30), and other wards (n=12/30).”

9) I think the abbreviations R, B and A do not improve the readability of the manuscript.

10) Include in the discussion that isolates with identical/related STR profiles are not necessarily the same strain, as they still might differ > 100 SNPs, as was for example shown for C. tropicalis (Microbiol Spectr. 2023 Jan 30;11(2):e0461822. doi: 10.1128/spectrum.04618-22.)

Author Response

Reviewer´s comment: The authors draw the conclusion that the gastrointestinal tract might constitute a potential reservoir of Candida spp potentially causing candidaemia as 14% of genotypes found in rectal swabs samples could also be detected in blood cultures. As different patient groups were used for this comparison, I do not agree with the interpretation of this data. In my opinion this is likely due to common environmental strains (in the study with vaginal swabs in general also much less clusters were found, so it is logical that as a consequence also less clusters with both vaginal and blood isolates are found; this study also contained much less isolates). However, the authors also found that “a total of 41/754 patients also had the same Candida spp isolated from blood cultures (n=18), intra-abdominal samples (n=20), or both (n=3). From these 41 patients with Candida spp in rectal swabs and isolates from blood (n=21) and/or intra-abdominal samples (n=23), 17/21 (80.9%) and 18/23 (78.3%) patients had genotypes matching the rectal swab and their blood cultures or their intra-abdominal samples, respectively.” I think this is much stronger evidence that “the gastrointestinal tract might constitute a potential reservoir of Candida spp potentially causing candidaemia”. These observed matches are not based on coincidence as between patients this number is much lower (14% on average). I think the manuscript should focus on the latter finding and use the other data as a “control group”.

Authors´ reply: We have reported in previous studies the presence of clusters involving patients epidemiologically unrelated, and sometimes those genotypes involve patients admitted at different hospitals, what we call a widespread genotype (Guinea et al., doi:10.3389/fcimb.2020.00166; Diaz-Garcia et al., doi:10.3390/jof8111228). A potential explanation to this observation is the fact that some genotypes might be more commonly present in the microbiota than others and are, therefore, more prone to cause candidemia. Our study was mainly designed to check if genotypes in the rectum can be also detected causing candidemia, and some of them fulfilled such hypothesis. This reviewer, and some others as well, has proposed a more detailed analysis of patients with available isolates from rectal swabs, blood cultures, and intra-abdominal samples. We agree with such a proposal, and we added such analysis at the beginning of the Results section, given the preponderance of such demand, and the fact that such group of patients sets an example of we are trying to prove. Hope the current draft is more satisfactory to reviewers.

Reviewer´s comment: Raw STR data should be provided as supplementary data.

Authors´ reply: Raw STR data are now included as supplementary material.

Reviewer´s comment: Table 1 and 2: Include the number of patients in both tables. Without this information, it is hard to understand the percentages at the bottom of the tables.

Authors´ reply: As we stated in M&M section (2.1), we studied one rectal swab sample per patient and species. However, we included such information in the tables.

Reviewer´s comment: Lines 118-120: Include description regarding isolates from database (blood cultures (n=650) or intra-abdominal samples (n=150)) in M&M. Also the structure of this paragraph should be improved. First describe Table 2 and then compare R, B and A.

Authors´ reply: We agree with the reviewer’ comment, and we have clarified this section.

Reviewer´s comment: Figure 1/lines 138-141; description colors and numbers should be in Figure legend. Improve explanation of Figure 1; in lines 126-129 and 142-144 is referred to Figure 1, but I cannot find these percentages in this figure.

Authors´ reply: We agree with the reviewer that the explanation in Figure 1 is difficult to follow. The legend to the figure had merged with the subsequent text due to the template format. Now it has been amended. To clarify the explanation to the figure, we have added the overall data and those broken down by species.

Reviewer´s comment: Figure 2/lines 154-155: description colors and numbers should be in Figure legend.

Authors´ reply: The legend to the figure had merged with the subsequent text due to the template format. Now it has been amended.

Reviewer´s comment: Figure 3/lines 188-193: description colors and numbers should be in Figure legend. Indicate what the number within the symbols represent (day of month?). Dotted lines are shown in different thickness. Is there any difference?

Authors´ reply: The number indicates the day of month, and we added this information in the figure caption. There are no differences in the thickness of the dotted lines; dotted lines indicate clonally related genotypes that we stated as: “In a given patient, a symbol shape indicates matches (identical genotype) and dotted edges depict clonally related genotypes”.

Reviewer´s comment: Lines 59-62: The sentence “A total 59 of 41/754 patients also had the same Candida spp isolated from blood cultures (n=18), intra-60 abdominal samples (n=20), or both (n=3).” should be in the Results section (line 162). Rephrase M&M like “In patients in whom the same Candida spp was isolated from gastrointestinal tract and blood and/or abdomen, we studied available additional 61 rectal swabs, blood cultures, and intra-abdominal isolates.”

Authors´ reply: Done as suggested.

Reviewer´s comment: Lines 111-114: I think this sentence “Of the 236 patients involved in clusters, 30 (12.7%) were involved in 10 clusters (2-5 patients per cluster) and were epidemiologically linked and admitted to intensive care (n=12/30), internal medicine (n=6/30), and other wards (n=12/30).” should read as “Of the 236 patients involved in clusters, 30 (12.7%) patients, involved in 10 clusters (2-5 patients per cluster), were epidemiologically linked and admitted to intensive care (n=12/30), internal medicine (n=6/30), and other wards (n=12/30).”

Authors´ reply: We appreciate the reviewer’ comment, and we have modified the sentence accordingly.

Reviewer´s comment: I think the abbreviations R, B and A do not improve the readability of the manuscript.

Authors´ reply: We have expanded the abbreviations to make the results easier to understand.

Reviewer´s comment: Include in the discussion that isolates with identical/related STR profiles are not necessarily the same strain, as they still might differ > 100 SNPs, as was for example shown for C. tropicalis (Microbiol Spectr. 2023 Jan 30;11(2):e0461822. doi: 10.1128/spectrum.04618-22.)

Authors´ reply: The limitation of microsatellite genotyping has now been included in the manuscript. We agree with the reviewer that microsatellite genotyping is less discriminative than whole genome sequencing. However, in the C. tropicalis study commented on by the reviewer, authors concluded than both techniques mostly correlated. We got the same observation in a previous study in with both tools were compared in C. albicans and C. parapsilosis.

Reviewer 2 Report

The manuscript by Mesquida et al. examined the levels of similarity among Candida isolates from rectal, blood, and intra-abdominal samples of hospitalized patients. Using species-specific microsatellite assays, they found that majority of the blood or intra-abdominal isolates were either identical or clonally related with the paired rectal isolates, thus claimed the gastrointestinal tract as a potential reservoir of invasive candidiasis. While the clinical strain collection is quite unique and valuable, some of the analyses in this manuscript are incomplete thus the arguments are less well supported.

Major issues

1.     Identical rectal and blood/intra-abdominal isolates from different patients only supports the circulation of a Candida strain in the hospital, not the source of the infection. The authors should focus on the 41 patients with paired rectal and blood/intra-abdominal isolates and add a table of similar format of table 3, with these 41 patients.

On the other hand, in table 4, the authors showed how many identical/clonally related pairs in the 41 patients with both rectal and blood/intra-abdominal isolates. However, to claim that the gastrointestinal tract is the reservoir of the invasive candidiasis, it is important to show the sequential order of these isolates, that is, the identical/clonally related rectal isolates are found prior to the blood or intra-abdominal isolates. In a murine model of invasive candidiasis, it has shown that intestinal Candida colonization occurs after intravenous infection with Candida albicans (PMID 35568028). It is possible that the patients acquire infections first (from medical devices or other sources), then have the fungal organisms colonize in their gut. In this case, even if the rectal isolates are identical with the blood isolates, it is not appropriate to pinpoint the gastrointestinal tract as the reservoir of the infection. The authors need to check the sample information of the 41 patients and re-analyze the data with timing information.

2.     The data presented in figure 2 is confusing. Where are the large number of isolates from? Why do the authors to compare them? These are totally missing in the manuscript. And again, if these isolates are from unrelated patients, it only supports that certain Candida isolates are circulating among the patients. This part of assay needs to be clarified. If it does not directly support the major conclusion of the manuscript, it can be removed or put in the supplementary data.

3.     One limitation of this sample collection is that the authors only analyzed one isolate per sample. This might explain in some of the patients with paired rectal and blood/intra-abdominal samples, the isolates from different samples are not related, since the potentially existed identical isolates were not sampled. The authors can include this in the discussion.

4.     In figure 3, what are the numbers labeled in the symbols? This longitudinal collection of rectal samples is very valuable, since it shows the same isolated could colonize in the patient for months and then lead to infection. The authors could look up the patient information more carefully for the potential risk factors of translocation or long-term colonization.

Minor issues:

1.     The format of table/figure legend is scrambled. Please use the same font across the legends.

2.     Line43-44, genotyping gut and blood Candida isolates by high resolution whole genome sequencing has been reported by PMID 31907459.

3.     If the data are available, please add a table of the characterization of the 41 patients with paired samples, which mainly includes the background conditions, antifungal drug exposure etc.

N/A

Author Response

Reviewer´s comment: Identical rectal and blood/intra-abdominal isolates from different patients only supports the circulation of a Candida strain in the hospital, not the source of the infection. The authors should focus on the 41 patients with paired rectal and blood/intra-abdominal isolates and add a table of similar format of table 3, with these 41 patients.

Authors´ reply: Genotyping results from the 36 patients with paired rectal and blood/intra-abdominal isolates have been added in table S1. We don’t think that identical rectal and blood/intra-abdominal genotypes from different patients indicated the circulation of the Candida genotypes in the hospitals, since rectal or intra-abdominal genotypes are not easy to spread from one patient to another. In contrast, patients with identical genotypes from blood are probably a consequence of nosocomial transmission. We reported in previous studies the presence of clusters involving patients epidemiologically unrelated, and sometimes those genotypes involve patients admitted at different hospitals, what we call a widespread genotype(Guinea et al., doi:10.3389/fcimb.2020.00166; Diaz-Garcia et al., doi:10.3390/jof8111228).

Reviewer´s comment: On the other hand, in table 4, the authors showed how many identical/clonally related pairs in the 41 patients with both rectal and blood/intra-abdominal isolates. However, to claim that the gastrointestinal tract is the reservoir of the invasive candidiasis, it is important to show the sequential order of these isolates, that is, the identical/clonally related rectal isolates are found prior to the blood or intra-abdominal isolates. In a murine model of invasive candidiasis, it has shown that intestinal Candida colonization occurs after intravenous infection with Candida albicans (PMID 35568028). It is possible that the patients acquire infections first (from medical devices or other sources), then have the fungal organisms colonize in their gut. In this case, even if the rectal isolates are identical with the blood isolates, it is not appropriate to pinpoint the gastrointestinal tract as the reservoir of the infection. The authors need to check the sample information of the 41 patients and re-analyze the data with timing information.

Authors´ reply: We have reviewed the clinical charts of the 36 patients with isolates from rectal swabs and intra-abdominal isolates and/or blood culture isolates (new Table 1 and Table S1).

Reviewer´s comment: The data presented in figure 2 is confusing. Where are the large number of isolates from? Why do the authors to compare them? These are totally missing in the manuscript. And again, if these isolates are from unrelated patients, it only supports that certain Candida isolates are circulating among the patients. This part of assay needs to be clarified. If it does not directly support the major conclusion of the manuscript, it can be removed or put in the supplementary data.

Authors´ reply: We have removed this Figure.

Reviewer´s comment: One limitation of this sample collection is that the authors only analyzed one isolate per sample. This might explain in some of the patients with paired rectal and blood/intra-abdominal samples, the isolates from different samples are not related, since the potentially existed identical isolates were not sampled. The authors can include this in the discussion.

Authors´ reply: This has now been added in the discussion section.

Reviewer´s comment: In figure 3, what are the numbers labelled in the symbols? This longitudinal collection of rectal samples is very valuable, since it shows the same isolated could colonize in the patient for months and then lead to infection. The authors could look up the patient information more carefully for the potential risk factors of translocation or long-term colonization.

Authors´ reply: The number indicates the day of moth and we added as request in figure legend. We have reviewed the clinical charts of the 36 patients with isolates from rectal swabs and intra-abdominal isolates and/or blood culture isolates (new Table 1 and Table S1).

Reviewer´s comment: The format of table/figure legend is scrambled. Please use the same font across the legends.

Authors´ reply: We appreciate the reviewer’ comment, and we have tidied it up.

Reviewer´s comment: Line43-44, genotyping gut and blood Candida isolates by high resolution whole genome sequencing has been reported by PMID 31907459.

Authors´ reply: The work with PMID 31907459 is a metagenomics work in which to determine the fungal load in fecal samples by means of 18S and ITS sequencing, and its relationship with the bacterial load/diversity. This paper only have 3 patients with candidemia isolates and simultaneous fecal samples in the same patient. In two of these patients, C. parapsilosis was isolated in both fecal and blood samples, and complete genome sequencing determined that it was the same isolate in only one of them. It is true that microsatellite markers are less discriminative than whole genome sequencing (this has been added to the discussion), but previous work by the group has compared microsatellite genotyping and whole genome sequencing, demonstrating that microsatellite genotyping it is a good tool for genotyping (Guinea et al., 2021, doi:10.1093/mmy/myab068).

Reviewer´s comment: If the data are available, please add a table of the characterization of the 41 patients with paired samples, which mainly includes the background conditions, antifungal drug exposure etc.

Authors´ reply: We have reviewed the clinical charts of the 36 patients with isolates from rectal swabs and intra-abdominal isolates and/or blood culture isolates (new Table 1 and Table S1).

Reviewer 3 Report

This manuscript by Mesquita et al aims to show that the gastrointestinal tract is a potential reservoir of C. albicans, C. parapsilosis and C. tropicalis invasive candidiasis (bloodstream infection or intra-abdominal infection). The isolates were genotyped by species-specific microsatellite method. Although the study presents an important topic, it is poorly written, and the data are at times poorly presented and/or miss-interpreted. The reviewer thinks there are multiple aims the authors are trying to evaluate, but these are not explicitly explained. The aims of the study need to be made clear, and the data pertaining to individual aims explicitly presented. The limitations of the genotype method, including potential for low discriminatory power in evaluating genetic relationship was not discussed. The manuscript needs a major rewrite and another cycle of review before it could be considered for publication

Specific comments:

1)      Abstract (lines 27-28): Under Results, the numbers are misleading. There are only 41 infections (18 blood, 20 intra-abdominal and 3 both sites), but numerous stool isolates over time. The numbers given are extremely confusing. Why are there so many stool isolates?

2)      Materials and Methods: Lines 56-67 – not sure why screening for multi-resistant bacteria was described here. Data were not given elsewhere

3)      Results:

a.       The numbers of patients and number of isolates per site (R, B, A) per patient per Candida species should be shown at the start of the results section.  

b.      This study is to look at the genetic relationship between the GI colonized stains and the disease causing strains. What are the justification for including 816 isolates, and not just the isolates from 41 patients with disease?

c.       Table 2. Where the 479 blood culture isolates come from? I thought there are only 18 patients with positive blood culture? Same questions for the abdominal sites

d.      Figure 1. Lines 138-139 should be presented as footnote for Figure 1 rather than in text. Why is inter-patient analysis done here? What does it have to do with the genetic of isolates from rectal swab and blood or intra-abdominal site? 4 patients (P1,3,4 and 6) had isolates recovered from the infected sites before the rectal isolate (some as far as ~2years). How can the authors be sure that the infected isolate did not precede the GI colonization isolate?

e.       Figure 3. Again, lines 188-200 belong to footnote of Fig 3. What do different shapes mean? Please define. How are genetically unrelated isolates denoted here?

4)      Discussion:

a.       The major limitation of the study is the typing method which might not be discriminatory enough to evaluate genetic relatedness.

Author Response

Reviewer´s comment: This manuscript by Mesquita et al aims to show that the gastrointestinal tract is a potential reservoir of C. albicans, C. parapsilosis and C. tropicalis invasive candidiasis (bloodstream infection or intra-abdominal infection). The isolates were genotyped by species-specific microsatellite method. Although the study presents an important topic, it is poorly written, and the data are at times poorly presented and/or miss-interpreted. The reviewer thinks there are multiple aims the authors are trying to evaluate, but these are not explicitly explained. The aims of the study need to be made clear, and the data pertaining to individual aims explicitly presented. The limitations of the genotype method, including potential for low discriminatory power in evaluating genetic relationship was not discussed. The manuscript needs a major rewrite and another cycle of review before it could be considered for publication.

Authors´ reply: We know that microsatellite markers are less discriminative than whole genome sequencing; however, it is a highly discriminative technique than mostly correlate with whole genome sequencing (de Groot et al., 2022, doi:10.1128/spectrum.02645-22; Guinea et al., 2021, doi:10.1093/mmy/myab068;  Spruijtenburg et al, 2023, doi:10.1128/spectrum.04618-22) We have added the microsatellite limitation to the discussion, and included the clinical characterization of patients. We have reorganized the results section and hope that the manuscript is more understandable now.

Reviewer´s comment: Abstract (lines 27-28): Under Results, the numbers are misleading. There are only 41 infections (18 blood, 20 intra-abdominal and 3 both sites), but numerous stool isolates over time. The numbers given are extremely confusing. Why are there so many stool isolates?

Authors´ reply: We have modified the results and hope that the manuscript is more understandable now.

Reviewer´s comment: Materials and Methods: Lines 56-67 – not sure why screening for multi-resistant bacteria was described here. Data were not given elsewhere.

Authors´ reply: We appreciate the reviewer’ comment, and we have done as request.

Reviewer´s comment: The numbers of patients and number of isolates per site (R, B, A) per patient per Candida species should be shown at the start of the results section.

Authors´ reply: We agree with the reviewer, and these data now are shown in the results section.

Reviewer´s comment: This study is to look at the genetic relationship between the GI colonized stains and the disease causing strains. What are the justification for including 816 isolates, and not just the isolates from 41 patients with disease?

Authors´ reply: We have been genotyping isolates from blood and other locations for more than 10 years and we have observed that genotypes in colonizing or vaginal samples rarely cause candidemia. In contrast, we found identical genotypes in intra-abdominal and blood samples. This study tried to decipher if genotypes present in the rectum could also be causing invasion. The presence of clusters involving patients epidemiologically unrelated is not new as we previously reported, sometimes as widespread genotypes grouping patients from different cities or countries. Therefore, we hypothesize that the presence of identical genotypes in rectal swabs or blood cultures from different patients might pinpoint some saprophytic genotypes with the ability of cause invasive disease.

Reviewer´s comment: Table 2. Where the 479 blood culture isolates come from? I thought there are only 18 patients with positive blood culture? Same questions for the abdominal sites.

Authors´ reply:  Isolates from blood and from the abdominal cavity were obtained from our genotyping database that has genotyped isolates since 2007 as we stated in the beginning of section 3.3.

Reviewer´s comment: Figure 1. Lines 138-139 should be presented as footnote for Figure 1 rather than in text. Why is inter-patient analysis done here? What does it have to do with the genetic of isolates from rectal swab and blood or intra-abdominal site? 4 patients (P1,3,4 and 6) had isolates recovered from the infected sites before the rectal isolate (some as far as ~2years). How can the authors be sure that the infected isolate did not precede the GI colonization isolate?

Authors´ reply: We appreciate the reviewer’ comment, and we have tidied it up.

Reviewer´s comment: Figure 3. Again, lines 188-200 belong to footnote of Fig 3. What do different shapes mean? Please define. How are genetically unrelated isolates denoted here?

Authors´ reply: We appreciate the reviewer’ comment. We have corrected the footnote for Figure 3. We appreciate the reviewer’ comment, and we have tidied it up. Different shapes mean different genotypes in a given patients, unrelated genotypes were denoted with different shapes without edge.

Reviewer´s comment: Discussion: The major limitation of the study is the typing method which might not be discriminatory enough to evaluate genetic relatedness.

Authors´ reply: We have added this limitation to the discussion.

Reviewer 4 Report

Ref: jof-2379855

Title:  The gastrointestinal tract is pinpointed as a reservoir of Candida albicans, Candida parapsilosis and Candida tropicalis genotypes found in blood and intra-abdominal samples

Journal: JoF

Review:

Background

The main objective of this study was to investigate if gastrointestinal tract can be a source of C. albicans, C. parapsilosis, and C. tropicalis for candidemia and intra-abdominal infections. By the use of microsatellite to detect the genotypes from samples collected by retal swab the authors identified several genotypes from different host niche, however part of the genotypes were common among them, indicating the gastrointestinal tract has a potential to disseminate Candida sp. to other parts of human body.

Minor concerns

The authors present interesting and highly impactful results for hospitalized patients. some points must be addressed:

1.                  It would be of great interest to readers if data related to patients such as age, sex, morbidity, clinical condition (if the patient is covid-19 positive, for example) could be correlated with the genotypes of Candida sp. found throughout the work.

2.                  Among the Candida species identified, in none of the samples was Candida auris found? A species of great concern in hospital environments?

3.                  Are the virulence phenotypes of Candida found exclusively in rectal specimens different from those found in more than one specimen (R+B+A or R+B or R+A)?

4.                  Authors must include in materials and methods how they extracted genomic DNA from the isolates.

5.                  Figure legends should be better written. For example in figure 1, lines 138-141 refer to the legend, the same for figure 2 lines 154 to 159 and figure 3 - lines 188-193.

6.                  In figure 3, what does the number inside the shape mean? It is not clear. Still in figure 3, it is indicating that patient 5 has 4 isolates, but in the figure we only have 3 isolates represented by the shapes.

7.                  It is highly recommended that authors improve the results section in order to make it more accessible to readers.

8.                   Check for self-plagiarism, on the section Microsatellite Genotyping – Lines 63-83

Genotyping was conducted as previously reported (7, 13). Species-specific microsat-64 ellite markers were used to genotype isolates of C. albicans (CDC3, EF3, HIS3 CAI, CAIII, 65 and CAVI) (14, 15), C. parapsilosis (CP1, CP4a, CP6, and B) (16, 17), and C. tropicalis (Ctrm1, 66 Ctrm10, Ctrm12, Ctrm21, Ctrm24, and Ctrm28) (18). Capillary electrophoresis using the 67 ABI 3730XL analyser (Applied Biosystems-Life Technologies Corporation, Carlsbad, CA, 68 USA) was carried out and electropherograms were analysed with GeneMapper v.4.0 software (Applied Biosystems-Life Technologies Corporation, Carlsbad, CA, USA). A molecularly identified control strain from each species was used in each run to ensure size accuracy and avoid run-to-run variations. The allele results were converted to binary data by scoring the presence or absence of each allele. Genetic relationships between genotypes were examined by constructing a minimum spanning tree (BioNumerics version 7.6, Ap-74 plied Maths, Sint-Martens-Latem, Belgium). Genotypes were encoded as CA-X (C. albi-75 cans), CP-X (C. parapsilosis), and CT-X (C. tropicalis), where X represents the internal code 76 of the genotype in our collection.

Paper: J. Fungi 2022, 8, 1228. https://doi.org/10.3390/jof8111228

2.2. Microsatellite Genotyping

Species-specific microsatellite markers were used to genotype isolates of C. albicans (CDC3, EF3, HIS3 CAI, CAIII, and CAVI) [19,20], C. parapsilosis (CP1, CP4a, CP6, and B) [21,22], and C. tropicalis (Ctrm1, Ctrm10, Ctrm12, Ctrm21, Ctrm24, and Ctrm28) [23]. Capillary electrophoresis using the ABI 3130xl (Applied Biosystems-Life Technologies Corporation, Carlsbad, CA, USA) analyzer was performed on the PCR products, and electropherograms were analyzed with the GeneMapper v.4.0 software (Applied Biosystems-Life Technologies Corporation, Carlsbad, CA, USA). A molecularly identified control strain from each species was used in each run to ensure size accuracy and avoid run-to-run variations. The allele results were converted to binary data by scoring the presence or absence of each allele. The data were treated as categorical, and the genetic relationship between genotypes was examined by constructing a minimum spanning tree (BioNumerics version 7.6, Applied Maths, Sint-Martens-Latem, Belgium). The isolates were considered to have identical genotypes when they presented the same alleles at all loci. Different genotypes were encoded as follows: CA-X (C. albicans), CP-X (C. parapsilosis), and CT-X (C. tropicalis), X representing the internal code of the genotype in our collection.

Author Response

Reviewer´s comment: It would be of great interest to readers if data related to patients such as age, sex, morbidity, clinical condition (if the patient is covid-19 positive, for example) could be correlated with the genotypes of Candida sp. found throughout the work.

Authors´ reply: We have reviewed the clinical charts of the 36 patients with isolates from rectal swabs and intra-abdominal isolates and/or blood culture isolates (new Table 1 and Table S1).

Reviewer´s comment: Among the Candida species identified, in none of the samples was Candida auris found? A species of great concern in hospital environments?

Authors´ reply: We did not find any Candida auris isolates in rectal swabs samples, blood cultures or abdominal samples in our hospital. Anyway this species was out of the scope of this study.

Reviewer´s comment: Are the virulence phenotypes of Candida found exclusively in rectal specimens different from those found in more than one specimen (R+B+A or R+B or R+A)? Authors´ reply: The reviewer has raised an interesting question, however we did not assess the virulence of the isolates.

Reviewer´s comment: Authors must include in materials and methods how they extracted genomic DNA from the isolates.

Authors´ reply: We performed microsatellites genotyping directly from cultures avoiding the DNA extraction. This information has been added to M&M section.

Reviewer´s comment: Figure legends should be better written. For example in figure 1, lines 138-141 refer to the legend, the same for figure 2 lines 154 to 159 and figure 3 - lines 188-193.

Authors´ reply: Done as request. All figure legends were poorly typed.

Reviewer´s comment: In figure 3, what does the number inside the shape mean? It is not clear. Still in figure 3, it is indicating that patient 5 has 4 isolates, but in the figure we only have 3 isolates represented by the shapes.

Authors´ reply: The number within the symbols indicates the day of sample collection. We added this information and the errors in the figure have been corrected.

Reviewer´s comment: It is highly recommended that authors improve the results section in order to make it more accessible to readers.

Authors´ reply: We have modified the results and hope that the manuscript is more understandable now.

Reviewer´s comment: Check for self-plagiarism, on the section Microsatellite Genotyping – Lines 63-83.

Authors´ reply: Out methodology is common to all our previous studies. Sometimes we have decided to give a brief overview of the method and references where details are available. However, reviewers commonly ask for more information, and then we need to write down our available procedure.

Round 2

Reviewer 1 Report

The first two lines of the abstract were deleted, leaving the abstract without any introduction sentences. This is not appropriate.

Author Response

Reviewer comment: The first two lines of the abstract were deleted, leaving the abstract without any introduction sentences. This is not appropriate.

Author’s reply: Done as suggested. However, we were forced to remove some sentences to fit the maximum 200 word count.

Reviewer 2 Report

Line 58: PMID 31907459 presented two patients, not one, with whole genome-matched intestinal colonized and blood C. parapsilosis isolates. Please revise the statement in the introduction.

Line 146-151: the authors stated there were two patients, patient P01 and P02, with three types of samples. However, in the revised version Figure 1, 1) there is no patient P02; 2) only patient P01 had three different colored symbols, the other patients all had just two colors, means none of them had three different types of samples. Which one is patient P02 as stated in the context? Please revise this figure and make sure the graphic presentation matches the context description.

In the discussion, only pointing out matched genotype in rectal and blood/intra-abdominal isolates from one patient is not sufficient to support the conclusion of “the gastrointestinal tract as a potential reservoir”. To determine a reservoir of infection, it is required to pinpoint the isolates at one location (such as GI tract) first, then the genotype identical isolates in blood or abdominal. The sequential order is the key factor. In fact, the authors do have the data to support their conclusion: in figure 1, there were six patients with rectal isolates sampled prior to the blood or abdominal isolates, all of the six patients had genotyped matched pairs. The data from these six patients are the key evidence to support the GI tract as the reservoir of infection. Please briefly modify the discussion.

n/a

Author Response

Reviewer comment: Line 58: PMID 31907459 presented two patients, not one, with whole genome-matched intestinal colonized and blood C. parapsilosis isolates. Please revise the statement in the introduction.

Author’s reply: Done as suggested.

Reviewer comment: Line 146-151: the authors stated there were two patients, patient P01 and P02, with three types of samples. However, in the revised version Figure 1, 1) there is no patient P02; 2) only patient P01 had three different colored symbols, the other patients all had just two colors, means none of them had three different types of samples. Which one is patient P02 as stated in the context? Please revise this figure and make sure the graphic presentation matches the context description.

Author’s reply: The figure only shows patients with multiple rectal swab samples studied. The data are correct, Patient P02 was only describe in Table S1 because there was only one rectal swab isolate available from him.

Reviewer comment: In the discussion, only pointing out matched genotype in rectal and blood/intra-abdominal isolates from one patient is not sufficient to support the conclusion of “the gastrointestinal tract as a potential reservoir”. To determine a reservoir of infection, it is required to pinpoint the isolates at one location (such as GI tract) first, then the genotype identical isolates in blood or abdominal. The sequential order is the key factor. In fact, the authors do have the data to support their conclusion: in figure 1, there were six patients with rectal isolates sampled prior to the blood or abdominal isolates, all of the six patients had genotyped matched pairs. The data from these six patients are the key evidence to support the GI tract as the reservoir of infection. Please briefly modify the discussion.

Author’s reply: As commented in the previous comment, Figure 1 only shows patients with multiple and available rectal swabs isolates. The reviewer is right and the smartest way to propose potential reservoirs is to detect the isolate there before than in blood/intra-abdominal samples. We have shown in Table S1 36 patients with paired rectal swabs and/or blood samples or intra-abdominal genotypes. In 18 of the patients, the rectal swab genotypes were detected prior to their detection in blood cultures/intra-abdominal samples. Given that our study was not addressed to detect rectal swabs genotypes in patients from their very moment of their hospital admission, we can not rule in/out that in the remaining 18 patients the rectal swab genotypes might’ve not been detectable prior to the development of the infection. However, in the cases in which the patients had multiple rectal swab genotypes over time, the genotypes were identical. We have added that as a limitation to the study.